# Reliable Fabrication of Graphene Nanostructure Based on e-Beam Irradiation of PMMA/Copper Composite Structure

**DOI:** 10.3390/ma14164634

**Published:** 2021-08-17

**Authors:** Kaixi Bi, Jiliang Mu, Wenping Geng, Linyu Mei, Siyuan Zhou, Yaokai Niu, Wenxiao Fu, Ligang Tan, Shuqi Han, Xiujian Chou

**Affiliations:** 1Key Laboratory of National Defense Science and Technology on Electronic Measurement, School of Instrument and Electronics, North University of China, Taiyuan 030051, China; bikaixi@nuc.edu.cn (K.B.); mujiliang@nuc.edu.cn (J.M.); wenpinggeng@nuc.edu.cn (W.G.); S2006103@st.nuc.edu.cn (S.Z.); 18834164264@163.com (S.H.); 2School of Mechanical Engineering, North University of China, Taiyuan 030051, China; meilinyu@nuc.edu.cn (L.M.); S2002107@st.nuc.edu.cn (Y.N.); feng062322@163.com (W.F.); 3Sichuan Jiuzhou Electric Group Co., Ltd., Mianyang 621000, China; tanlg@jezetek.cc

**Keywords:** graphene nanostructure, e-beam lithography, copper-catalyzed growth, local heating effect, in-situ synthesis

## Abstract

Graphene nanostructures are widely perceived as a promising material for fundamental components; their high-performance electronic properties offer the potential for the construction of graphene nanoelectronics. Numerous researchers have paid attention to the fabrication of graphene nanostructures, based on both top-down and bottom-up approaches. However, there are still some unavoidable challenges, such as smooth edges, uniform films without folds, and accurate dimension and location control. In this work, a direct writing method was reported for the in-situ preparation of a high-resolution graphene nanostructure of controllable size (the minimum feature size is about 15 nm), which combines the advantages of e-beam lithography and copper-catalyzed growth. By using the Fourier infrared absorption test, we found that the hydrogen and oxygen elements were disappearing due to knock-on displacement and the radiolysis effect. The graphene crystal is also formed via diffusion and the local heating effect between the e-beam and copper substrate, based on the Raman spectra test. This simple process for the in-situ synthesis of graphene nanostructures has many promising potential applications, including offering a way to make nanoelectrodes, NEMS cantilever resonant structures, nanophotonic devices and so on.

## 1. Introduction

Graphene nanostructures, forming a promising 2D nanomaterial with excellent properties, have been widely studied for electronic applications, sensing, biomedicine, and other fields [1,2,3]. Despite its numerous extraordinary properties for various applications, the biggest challenges existing for graphene applications are the reliable construction of a graphene nanostructure. Therefore, a great deal of effort has been directed toward fabricating a uniform graphene nanostructure without significantly affecting its basic features [4,5,6]. Nowadays, the engineering of graphene nanostructures is mainly based on top-down and bottom-up approaches. Although a micro/nano-meter graphene nanostructure can be acquired, the current fabrication methods are time-consuming, the manufacturing process is complicated, giving poor precision, and it is hard to integrate a functional graphene nanostructure into devices. As a result of various application requirements, reliable and high-precision preparation is becoming an important research topic.

Generally, graphene nanostructures can be produced via the top-down method, which mainly includes micromechanical cleavage, mask patterning and graphene etching. Among these processes, the mask patterns on the graphene surface need to be acquired first, through various forms of lithography technology, synthetic block copolymer (BCP) and other nanomaterial structures [7,8,9,10]. Then, oxygen plasma etching equipment would be employed to eliminate unmasked graphene through an oxidation reaction, and masked graphene is then retained as a regular nanostructure array. In the graphene pattern transfer process, resolution loss and the introduction of defects are inevitable. Compared with the top-down method, the bottom-up approach is another significant route for preparing graphene nanostructures [11,12,13,14,15], such as micromechanical cleavage methods, chemical vapor deposition (CVD) on a templated copper substrate, the self-assembly of organic molecules, graphene synthesis by ion-beam implantation, and so on. Although some graphene nanoribbons or nanodots can be prepared using the top-down approach, ensuring size and location control still has many difficulties.

Apart from these techniques, e-beam irradiation in-situ enables the growth of a functional graphene nanostructure in a unique high-precision preparation method, representing an advanced fabrication method in this field. Duan et al. acquired a graphene nanoribbon from ultrathin electrospun poly (methyl methacrylate) PMMA nanofibers via the e-beam irradiation process, in which the graphene sheets with a honeycomb hexagonal crystal structure can be prepared during continuous e-beam irradiation [16]. Chen et al. also fabricated a graphite electrode by irradiating PMMA film on a SiO_2_/Si substrate, to construct a high-resolution graphitized nanostructure with the help of the post-annealing process [17]. Furthermore, Andrey Turchanin et al. proposed a route based on the conversion of organic self-assembled monolayers for high-quality graphene nanostructures [18]. By adjusting production conditions, graphene nanostructure sheets can be altered. E-beam-induced in-situ fabrication of graphene nanostructure technology has gradually become the typical method for the high-resolution fabrication of graphene nanostructures. However, the high-energy e-beam is detrimental to high-quality graphene formation, due to the electronic scattering effect between organic materials and SiO_2_/Si substrate. Compared with an insulating SiO_2_/Si substrate, a copper substrate is useful to synthesize single or small layers of graphene because of its catalytic effect. Meanwhile, the excellent conductivity of copper substrate is also beneficial as a way to avoid the proximity effect during the e-beam lithography process, which is another point in favor of high-resolution graphene fabrication.

In this work, we adopted high-resolution PMMA organic resist for carbon-based precursors. Then, PMMA film on a copper substrate was selectively irradiated with a high-energy e-beam. By combining this with the subsequent high-temperature annealing process, various graphene nanostructures, including nanoribbon, dot arrays and other patterns, can be created. The minimum feature size of a graphene line is about 15 nm, and the irradiated products have a uniform and smooth morphological structure. The Fourier infrared absorption spectrum (FITR) test and Monte Carlo simulation also exhibit a molecular dynamic process between the incident e-beam and PMMA organic film. As a simple characterization tool, Raman spectroscopy is also used for characterizing the properties of graphene; the results were 1350 cm^−1^ (D-mode), 1583 cm^−1^ (G-mode) and 2680 cm^−1^ (2D-mode). In conclusion, a nanoscale high-quality graphene nanostructure can be reliably fabricated with the help of a high-precision e-beam fabrication method using a high-temperature copper-catalyzed graphene growth approach.

## 2. Experimental Section

### 2.1. Fabrication of the Graphene Nanostructure

Standard PMMA resist, with 950 k molecular weight and a concentration of 6% in chlorobenzene, was bought from EM RESIST LTD. A Zeiss scanning electron microscope (SEM) equipped with a pattern generator from the Raith company was used for high-resolution e-beam lithography. The irradiation conditions between the e-beam and PMMA organic film were: 10 kV to 30 kV EHT voltage, 5000 μC/cm^2^ to 12,000 μC/cm^2^ exposure dose, and 320 pA beam current. The irradiated PMMA film was soaked in acetone solution for 120 s and then in deionized isopropyl alcohol for 60 s. Subsequently, the sample was put into a quartz tube at 900 °C for 20 min with a heating rate of 10 °C/min under vacuum conditions. The few-layers graphene nanostructure then formed on the copper surface with the temperature of the tube furnace cooling down to room temperature.

### 2.2. Morphology Characterization

An SEM (Zeiss SUPRA-55) was used to produce a two-dimensional image and revealed information about the irradiated graphene nanostructures. A sub-nanometer resolution of the sample was presented under a 10 kV accelerating voltage and ~310 pA beam current.

The sample roughness can be observed by using atomic force microscopy (AFM, MFP-3D Origin Plus, Asylum Research, Oxford, UK). The AFM test conditions were set to contact mode for clearer morphological characterization. The 50 μm × 50 μm AFM image was collected at a 1Hz scan rate in an air atmosphere with a silicon cantilever (AC240TS-R3, Asylum Research, USA).

### 2.3. Material Characterization

Raman characterization is a convenient and nondestructive testing technology for acquiring accurate information on molecular structure, material defects, surface dangling bonds, and so on. The Raman test was achieved with a low-powered 532-nm incident laser and a long integral time (10 s for one spectrum). In order to ensure the accuracy of the test data, 3~5 Raman spectra were collected from the same sample, avoiding accidental errors. All Raman tests were operated at room temperature, in an air atmosphere and dark-field mode. In the Fourier transform infrared spectroscopy (FTIR, TENSOR 27, Rheinstetten, Germany) test, the radiation emitted by a light source was converted to interference light by an interferometer, and an infrared spectrum of the absorption of irradiated PMMA can be obtained, based on Fourier transform analysis using a computer. Here, the spectra were recorded between 4000 and 1000 cm^−1^. The XRD (XRD-7000S/L, Kyoto, Japan) characterization was adopted to study the crystalline structure of the irradiated PMMA sample. The X-ray diffractometers were used based on the θ/θ continuous scanning mode. Other working parameters, such as the 2.0000 deg/min scanning speed, 0.0500-degree sample tilt, 0° to 90° scanning angle and 1.5 s adjustment, were set to meet the test requirements.

## 3. Results and Discussion

PMMA is a typical high-resolution positive/negative e-beam resist for nanostructure fabrication. It has many advantages in terms of long-lasting results and a good sticking ability to the target substrate surface, and it is cost-effective and highly reproducible. Moreover, it can be transformed into a graphite-like material under e-beam irradiation. Considering the aim of this paper is to construct a graphene nanostructure, the copper substrate was used for fabrication, based on the Cu-catalyzed graphene growth mechanism. As shown in Figure 1, the 0.4-μm copper film was prepared on a SiO_2_/Si substrate using magnetron sputtering equipment (step 1). Then, a PMMA layer of 50-nm thickness was spin-coated on the copper substrate (step 2). Subsequently, the Raith-150 two electron-beam lithography system was used for selective irradiation at a 30-kV acceleration voltage with an e-beam current of 320 pA. The PMMA was gradually driven into the carbonaceous material with an increasing e-beam irradiation dose (step 3). The graphic nanostructures could be obtained from the PMMA after soaking in acetone for two minutes. In order to further acquire high-quality graphene crystal film, the sample was then put into a tube furnace at 900 °C for 10 min. It would be useful to change the size thickness and properties of the produced graphene nanostructure with this in-situ process using an e-beam direct-writing and annealing step.

The radial energy density of incident electrons is closely related to organic PMMA film parameters, including resist thickness, surface roughness, EHT (extra-high tension) voltage, and so on. The PMMA film is 50-nm thick, for preparing ultrathin graphene. During the e-beam irradiation process, the incident electron-scattering effects, which mainly include forward-scattering electrons and back-scattering electrons, affecting the energy distribution in the PMMA film. In the electron-scattering process, the electrons that have interacted with PMMA valence electrons are the major factor that causes PMMA molecular structure changes. In addition, inelastic scattering between incoming electrons and atomic electrons in organic molecular conditions creates a large amount of energy transformed into heat, causing a locally high temperature for a chemical reaction. The higher local temperature is beneficial to driving copper-catalyzed graphene growth, based on a PMMA solid organic source [19,20].

Electron scattering in solids is a very complex physical problem. Generally, the Monte Carlo simulation (simulated using the CASINO software, Figure 2) is used for molecular dynamics simulation studies of the interaction between high-energy particles and PMMA organic thin films. 

With this simulation, the detailed motion trajectory and energy density distribution of the electron can be clearly calculated. By calculating the scattering process of each electron in the photoresist and recording and analyzing the energy deposition density in the photoresist, the physical and chemical change processes at different positions can be estimated. In this paper, we needed to identify graphene transformation and high-resolution patterns. The dimensional variation of the e-beam spot can be given empirically (i_beam_ = KBd^8/3^, where “i” is the beam current, “B” is a constant, “d” is the size of e-beam, “k” is parameter related to electron optical spherical aberration). According to the formula, the smaller beam spot size required a higher acceleration voltage, which is also useful for the molecular structure transformation of PMMA. Eventually, graphene-like sheets can be obtained from PMMA organic film via the removal of the oxygen-containing functional groups by the e-beam irradiation process.

In this study, a 2 × 2 irradiated PMMA array with a center-to-center distance of 30 μm was fabricated by the e-beam lithography process. The square sample was approximately 20 μm wide. By using optical characterization, including the Fourier infrared absorption test, Raman spectra and XRD spectra, we found that the hydrogen and oxygen elements were disappearing due to elastic scattering effects and inelastic scattering effects between the incoming electrons and organic film. The graphene crystal was also preliminarily formed by the diffusion and local heating catalytic effect between the e-beam and copper substrate.

We confirmed the changes in chemical elements and functional groups using FTIR spectra detection, which showed that the oxygen-containing groups gradually died out with the increasing irradiation doses. In the measurements, the interference light source was introduced into irradiated PMMA film for measuring the absorption spectra. The scanning was recorded across a 4000 and 1000 cm^−1^ scope. In the Fourier transform infrared spectroscopy (FTIR, TENSOR 27, Rheinstetten, Germany) test, the radiation emitted by a light source was converted to interference light by an interferometer, and an infrared spectrum of the absorption of irradiated PMMA can be obtained with a computer, based on Fourier transform analysis. Here, the spectra were recorded between 4000 and 1000 cm^−1^.

Figure 3b shows the changes in the irradiated PMMA sample with different irradiated processes. It is clear that the peaks of some chemical functional groups, i.e., hydroxyl C-OH (1416 cm^−1^), -OH (3410 cm^−1^) and -OH groups of dimeric COOH groups (2927 cm^−1^), were obvious in the initial stages. The signal intensity of the oxygen-containing groups gradually disappeared with the increase of the e-beam dose. By the 10,000 μC/cm^2^ doses of irradiation, the oxygen-containing groups had almost disappeared. The C=C (1662 cm^−1^) and -CH_2_- (2927 cm^−1^) chemical bond of the PMMA molecular formula also cracked under high-dose irradiation [21,22]. The FTIR test spectra implied that the cracking reaction was produced during a continuous e-beam irradiation process. All FITR absorption spectra have been tested on GaF substrate, due to its excellent transmittance in the ultraviolet-visible near-infrared optical frequency band, which is useful for studying the properties of the irradiated PMMA samples at all stages.

Raman characterization is widely used in structural optical characterization. Much of the work using Raman tests of graphitic materials was reported, and it has gradually become a convenient and nondestructive testing technology for samples. The molecular material information can be obtained by the interaction between a 532-nm incident laser and molecular structural vibration [23,24]. For the SiO_2_ substrate sample shown in Figure 3c, the D peak at 1350 cm^−1^ indicates that the sample has obvious defects and disordering. The G peak at 1580 cm^−1^ means that the sample has formed crystalline graphite. The D and G peaks may result from the charge-accumulation effect of the insulating SiO_2_ layer and limited carbon atom diffusion on the SiO_2_ surface. Different forms of SiO_2_ substrate and copper substrate have excellent electrical conductivity and catalytic effects. For the copper substrate sample (Figure 3c), the G band (1582 cm^−1^) in the spectrum is coming from the first-order scattering process of graphene crystal. The 2D and D bands (~2700 cm^−1^ and ~1350 cm^−1^) originate from a second-order scattering process, corresponding to two or one phonons near the K point and one defect. The prominent features that are shown, including the G band, D band and 2D band in Raman spectra, are compelling evidence to verify the existence of graphene, although it still has some defects because of the D peak and wide non-Gaussian distribution 2D peak. The value of I_D_/I_G_ ≈0.4 implies that the lower defects were produced on the copper substrate. The FWHM of the 2D peak is about 123 cm^−1^, which is much higher than for 25–60 cm^−1^ of multilayer graphene. The annealing process is necessary for acquiring a single- or few-layer graphene sample. The statistical evaluation of the graphene parameters as 2D width, D/G intensity ratio and the 2D/G intensity ratio has also been provided in the Appendix A for discussion (Appendix A).

During the XRD measurement process, the samples were irradiated with Cu Kα radiation (λ = 0.1545 nm). The 3c range and step sizes were 10–80° and 0.02°, respectively. The measurement time for the sample was 9 min. There is no obvious peak in the XRD spectrum of the pure PMMA sample. After e-beam irradiation, a weak and broad diffraction peak appeared at 21.3°, which can be attributed to the (002) reflection of graphene nanosheets. This peak indicates that the first minimum graphene/graphite structure was formed, which is consistent with the hexagonal structure [24]. It also indicates that PMMA was successfully transferred to the multilayer graphene or graphite material.

For further optimizing the growth process and the ultimate graphene nanostructure properties, the high-temperature annealing process plays an essential role. Benefiting from the low-carbon-solubility metal, the high-quality monolayer or few-layer graphene nanostructure can be grown on the copper surface, due to the mechanism of dissolution and the precipitation of copper. The topographic AFM images of graphene before and after the annealing process are shown in Figure 4a,b. The thickness of the irradiated PMMA decreased from ~40 nm to ~1.08 nm. The results imply that an annealing process at a high temperature is conducive to removing the remaining carbon-based materials. The edge of the graphene was also not attached to the substrate surface completely in Figure 4b. The surface crimp phenomenon between graphene and copper substrate may result from this disengagement from the surface. The height values are measured before and after annealing at 900 °C in a vacuum at 40 nm and 1.08 nm, respectively. The mechanism involved in the removal of graphene residues may be the high-temperature dissolution and catalytic properties of the copper substrate [25].

As we know, high-temperature annealing can drive material cleavage and recombination, which is a useful technology to repair defects and remove impurities. In this work, by combining chemical reduction reaction with thermal annealing process, the quality of graphene has been further improved. At this point, the irradiated PMMA sample was put in a high-temperature furnace in a vacuum, increasing graphene crystallization and burning off the residual PMMA. Figure 4c details the Raman peak changes in irradiated PMMA at the different stages. No characteristic Raman peaks were shown in the pure PMMA film without any annealing process. Three peaks (D, G and 2D peaks) were produced for the irradiated PMMA. The details can be seen in Figure 3c. When a series of annealing processes were introduced, the D peak of the sample gradually disappeared, which implies that improved deoxygenation and defect restoration was successful. A narrow and Gaussian-distributed 2D peak illustrated that few-layers graphene was achieved with the help of the annealing process. There are two mechanisms in graphene synthesis on a copper substrate during the annealing process: carbon precipitation and surface diffusion. The carbon atoms are firstly dissolved and precipitated on the copper surface, then some carbon atoms will be desorbed from the surface of the metal layer under the low ambient pressure. As a result of the limited carbon source, the number of graphene layers was gradually reduced with increasing annealing time, which led to a narrow and Gaussian-distribution 2D peak in the Raman test [26].

In addition to Raman characterization for all steps, a photoluminescence characterization was also adopted. As we know, pure PMMA film and graphene cannot produce PL signals because of their zero bandgaps. However, the graphene with doping and defects may have PL singles [27,28]. Figure 4d compares PL for pure PMMA, irradiated PMMA, samples with annealing for 10/20 min, and pure graphene film. It is obvious that pure PMMA and graphene have no photoluminescence signals. Irradiated PMMA and the annealing samples have a similar photoluminescence peak. An obvious visible peak with a long infrared emission tail was also observed in the PL spectra. The 45-nm-thick irradiated PMMA carbonaceous material gave rise to inter-flake relaxation pathways for free excitons. A redshift trend of emission photoluminescence spectra was produced, due to a strong interlayer coupling effect. When the annealing process was introduced, the irradiated PMMA carbonaceous material tended to be a thin graphene film containing -OH and -COOH chemical bonds.

As a direct writing technology, it is necessary to test the fabrication limit of this nanostructure. Figure 5 shows SEM and AFM images of graphene nanoribbon arrays on common Si/SiO_2_ substrate and copper substrate. In Figure 5a, we have designed and fabricated different sizes of graphene ribbon from the left to the right side on the copper substrate, and the feature size is 50 nm. The uniform and smooth morphology of graphene indicate a reliable fabrication process. However, the fabrication of a smaller linewidth graphene nanostructure is limited by the e-beam scattering effect on a rough copper surface, prepared by magnetron sputtering technology. Compared with the copper substrate, the SiO_2_/Si substrate has a flatter surface. We can clearly see that the widths of the nanoribbons prepared by e-beam lithography are close to the same size (~15 nm) in both SEM and AFM images (Figure 5c,d). The smoother substrate surface means achieving a finer graphene ribbon. A smoother copper surface is needed to improve the processing accuracy of a graphene ribbon.

## 4. Conclusions

As has been shown in this paper, we reviewed the fabrication of graphene nanostructures, as explained in the Introduction section. Facing the challenge of the reliable fabrication of a high-resolution and high-quality graphene nanostructure, we combined the advantages of the e-beam irradiation effect and the copper catalytic effect. The graphene array with a line width of 50 nm for the 150 nm pitch was fabricated by putting forward a simple one-step process for directly obtaining patterned carbonaceous nanostructures, and a high-temperature annealing process. The acquired graphene sample has an obvious Raman characteristic peak, such as the D peak (~1350 cm^−1^), G peak (~1582 cm^−1^) and 2D peak (~2700 cm^−1^). Other optical characterizations, including XRD and FTIR spectra, also clearly exhibited the fabrication mechanism. In addition, e-beam lithography has played an important role in the fabrication of graphene nanostructures on the copper substrate. However, it is hard to further reduce the minimum feature size of the graphene nanostructure due to the disorderly scattering of electrons resulting from the rough surface of the copper film as prepared by magnetron sputtering technology. In short, the work provides some new ideas and possibilities for the preparation of high-resolution and high-quality graphene nanostructures, which will further the development of carbon-based electronics.

## Figures and Tables

**Figure 1 materials-14-04634-f001:**
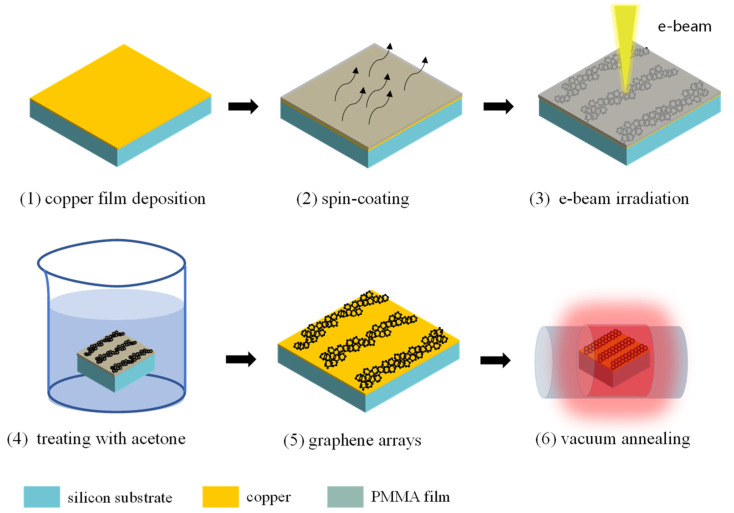
The process flow diagram of the e-beam lithographic process for the fabrication of graphene nanostructures on copper substrate. Low-quality graphene was formed at step 3, and the high-quality graphene at step 6.

**Figure 2 materials-14-04634-f002:**
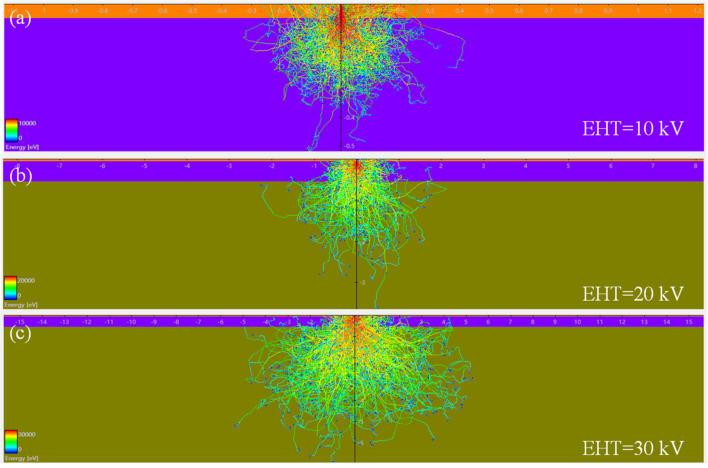
Monte Carlo simulations of the electron energy distribution into the PMMA film with an e-beam energy of 10, 20 and 30 keV. (**a**) EHT = 10 kV; (**b**) EHT = 20 kV; (**c**) EHT = 30 kV. The higher-energy electrons are highlighted in red, the lower-energy electrons in yellow and green.

**Figure 3 materials-14-04634-f003:**
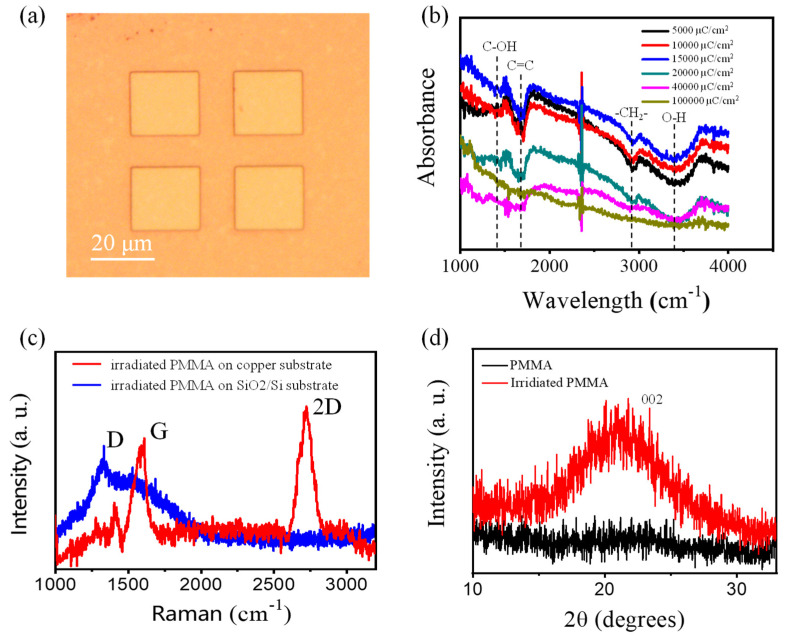
(**a**) Microscopy image of the e-beam-irradiated PMMA sample. (**b**) Fourier transform infrared (FTIR) absorption optical curves of PMMA film, irradiated with various exposure doses. (**c**) Raman spectra of irradiated PMMA film on copper substrate. (**d**) X-ray diffraction analysis of PMMA and irradiated PMMA film.

**Figure 4 materials-14-04634-f004:**
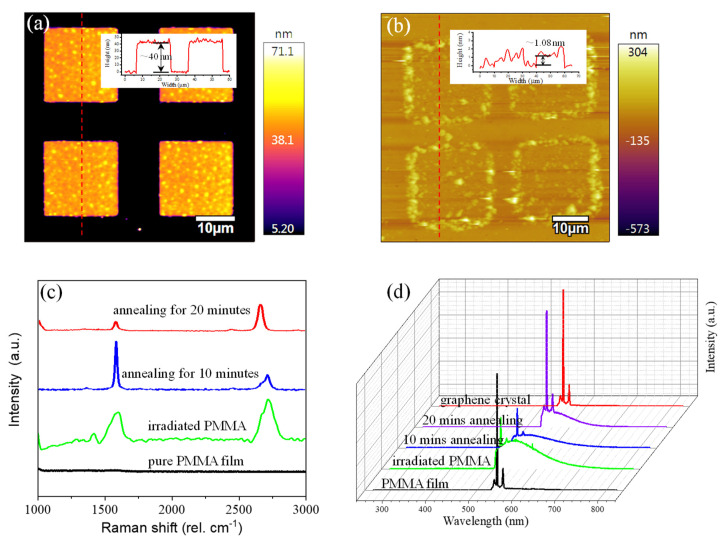
AFM view of the sample (**a**) before, and (**b**) after annealing at 900 °C in a vacuum atmosphere. (**c**,**d**) Raman and photoluminescent spectra of the sample at the different preparation stages.

**Figure 5 materials-14-04634-f005:**
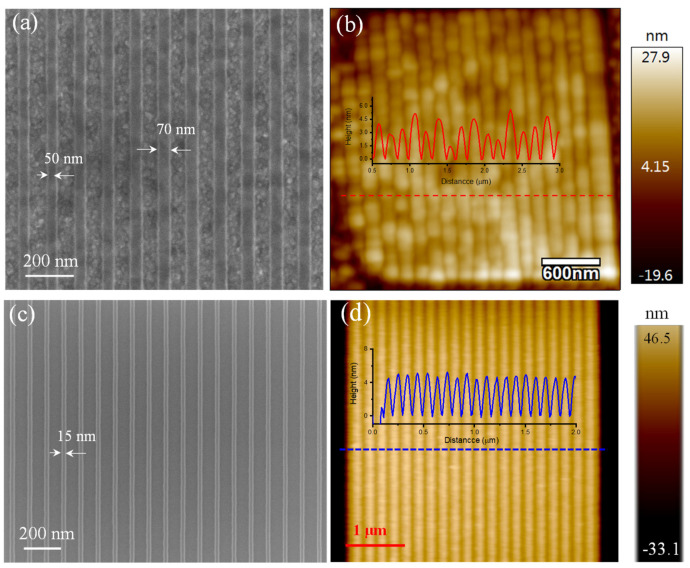
Graphene nanoribbons. (**a**,**c**) SEM images of graphene nanoribbons fabricated by e-beam direct writing on copper film substrate and SiO_2_/Si substrate, respectively. (**b**,**d**) AFM images of graphene nanoribbons on copper film substrate and SiO_2_/Si substrate, respectively.

## Data Availability

Not applicable.

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
