# Peer review of "Reliable Fabrication of Graphene Nanostructure Based on e-Beam Irradiation of PMMA/Copper Composite Structure"

_materials, 2021, doi:10.3390/ma14164634_

Round 1
Reviewer 1 Report
This work presents a synthesis method of graphene nanopatterns based on e-beam irradiation and thermal annealing of PMMA layers on Cu substrates. After polishing it based on the following minor critiques, this paper may be fit for publication in Materials. Therefore, I would like to recommend that this paper needs minor revision to be published in Materials.
1. Differences and advances of this work compared to refs. 16, 17, and 18 might be unclearly presented in the introduction. I think that addition of Cu substrates is the major difference/advance in this work. I would like to recommend you to add this point in the introduction, including pros and cons of the Cu layer addition.
2. Therefore, I believe that Raman spectra in Fig. 3c are important because they show the comparison between e-beam-irradiated PMMA on Cu and e-beam-irradiated PMMA on SiO2. Could you please add the comparison between Raman spectra of e-beam-irradiated/annealed PMMA on Cu (You already have this in Fig.4c.) and e-beam-irradiated/annealed PMMA on SiO2, and also add related discussions? And, in Raman spec. measurements, how many samples were measured? How many measurements were performed? Could you please clearly present these numbers in the main manuscript, and also present additional Raman data in the supplementary information?
3. Figure subpanel indicators and captions (Fig. 3b, 3c …) are incorrectly presented in Fig. 3 and related descriptions.
4. Could you please consider updating Figure 1? I am not sure whether graphene is formed at the step 3 or step 6? The process might be the formation of low-quality graphene at the step 3 and the re-crystallization to high-quality graphene at the step 6. In any case, the current form of Figure 1 might be unclear to readers.
5. Line 47: down-up -> bottom up. Does the micromechanical cleavage method belong to the bottom-up approach?
6. Line 236: Cleavage -> cleavage, Line 259: tend -> tends.
7. If you can provide TEM images (and/or SAED patterns) of synthesized graphene (by transferring it based on attaching a TEM grid and etching Cu), it should be very good. But, I know it is very difficult, so this comment is just a recommendation, not a mandatory task.
Author Response
Referee #1
Comments:
“This work presents a synthesis method of graphene nanopatterns based on e-beam irradiation and thermal annealing of PMMA layers on Cu substrates. After polishing it based on the following minor critiques, this paper may be fit for publication in Materials. Therefore, I would like to recommend that this paper needs minor revision to be published in Materials.”
Reply: We appreciate the referee’s positive comments on our work.
“1) Differences and advances of this work compared to refs. 16, 17, and 18 might be unclearly presented in the introduction. I think that addition of Cu substrates is the major difference/advance in this work. I would like to recommend you to add this point in the introduction, including pros and cons of the Cu layer addition.”
Reply:
We appreciate the referee’s suggestion. The highted text “E-beam induced in-situ fabrication of graphene nanostructures technology gradually become typical method for high-resolution fabrication of graphene nanostructures. However, high-energy e-beam is harmful to high-quality graphene formation due to its electronic scattering effect between organic materials and SiO2/Si substrate. Comparing with insulating SiO2/Si substrate, copper substrate is useful to synthesize single or small layers of graphene because of its catalytic effect. Meanwhile, the excellent conductivity of copper substrate is also beneficial to avoid proximity effect during e-beam lithography process, which is in favor of high-resolution graphene fabrication.” was added in introduction to disscuss the pros and cons of the Cu layer addition.
“2) Therefore, I believe that Raman spectra in Fig. 3c are important because they show the comparison between e-beam-irradiated PMMA on Cu and e-beam-irradiated PMMA on SiO2. Could you please add the comparison between Raman spectra of e-beam-irradiated/annealed PMMA on Cu (You already have this in Fig.4c.) and e-beam-irradiated/annealed PMMA on SiO2, and also add related discussions? And, in Raman spec. measurements, how many samples were measured? How many measurements were performed? Could you please clearly present these numbers in the main manuscript, and also present additional Raman data in the supplementary information?”
Reply:
Thanks for your kind advice.
Thanks for your review and we really appreciate for your time and patience. Firstly, we apologize for our vague statement of figure 3c. Secondly, we have already provided e-beam-irradiated PMMA on copper and e-beam-irradiated PMMA on SiO2 substrate in figure 3c. And we have modified the figure 3c according to the reviewer’s feedback. Besides that, we also add some discussion about figure 3c in results and discussion section.
We added “For SiO2 substrate sample (figure 3c), the D peak at 1350 cm-1 indicates that the sample has obvious defects and disordering. The G peak at 1580 cm-1 means that the sample has formed a crystalline graphite. The reason for D and G peak may result from a charge-accumulation effect of insulting SiO2 layer and limited carbon atoms diffusion on SiO2 surface. Different form SiO2 substrate, copper substrate has properties of excellent electrical conductivity and catalytic effect.” in results and discussion section of manuscript (figure 3c).
Commonly, all Raman spectra were measured for three samples at least. And each sample was tested five times at different areas of sample. We have provided additional Raman mapping and Raman spectra in supplementary information for further discussion.
“3) Figure subpanel indicators and captions (Fig. 3b, 3c …) are incorrectly presented in Fig. 3 and related descriptions.”
Reply:
Thanks for your careful reviewing. We have carefully checked the manuscript again. And we are sincerely sorry for the incorrectly presented in Fig. 3 and related descriptions. According to your suggestion, we have made changed as below:
We have changed descriptions about Fig. 3b from “Raman spectra of irradiated PMMA film on copper substrate” to “X-Ray diffraction analysis of PMMA and irradiated PMMA film”. Meanwhile, changed descriptions about Fig. 3c from “X-Ray diffraction analysis of PMMA and irradiated PMMA film” to “Raman spectra of irradiated PMMA film on copper substrate”.
Figure 3 (a) Microscopy image of the e-beam irradiated PMMA sample. (b) X-Ray diffraction analysis of PMMA and irradiated PMMA film. (c) Raman spectra of irradiated PMMA film on copper substrate. (d) Fourier transform infrared (FTIR) absorption optical curves of PMMA film irradiated with various exposure dose.
“4) Could you please consider updating Figure 1? I am not sure whether graphene is formed at the step 3 or step 6? The process might be the formation of low-quality graphene at the step 3 and the re-crystallization to high-quality graphene at the step 6. In any case, the current form of Figure 1 might be unclear to readers.”
Reply:
We would like to expend our sincere apologies for the vaguely presentation in the manuscript. According to your suggestion, we have modified figure1 and updated new version. Then we added highlighted explanatory text in figure 1 note to display the graphene formation in step 3 and 6.
Figure 1 The process flow diagram of the process of e-beam lithographic process for fabrication of graphene nanostructures on copper substrate. Low-quality graphene formed at the step 3 and the high-quality graphene at the step 6.
“5) Line 47: down-up -> bottom up. Does the micromechanical cleavage method belong to the bottom-up approach?”
Reply:
Thanks for your careful and professional review. According to your suggestion, we have changed them as below:
We change “…the down-up approach is another significant route…” to “…the bottom-up approach is another significant route…” on line 47.
Top-down method refers to the reduction of a bulk material to nanometric scale particles. Mechanical exfoliation was first described in 2008, which is also a top-down approach that requires mechanical energy to exfoliate graphite (Poniatowska, A. et al. Production and properties of top-down and bottom-up graphene oxide. J. Colloids & Surfaces A: Phys.2019:315-324; Neeraj, K. et al. Top-down synthesis of graphene: A comprehensive review. J. Flat Chem. 2021:100224).
Then we change “Commonly, the graphene nanostructures can be produced via top-down method mainly including mask patterning and graphene etching.” to “Commonly, the graphene nanostructures can be produced via top-down method mainly including micromechanical cleavage, mask patterning and graphene etching.” in introduction section.
“6) Line 236: Cleavage -> cleavage, Line 259: tend -> tends.”
Reply:
Thanks for your careful review. According to your suggestion, we have changed them as below:
We change “…such as micromechanical Cleavage methods…” to “…such as micromechanical cleavage methods…” on line 236.
We change “…the irradiated PMMA carbonaceous material tend to be a thin graphene film…” to “…the irradiated PMMA carbonaceous material tends to be a thin graphene film…” on line 259.
“7) If you can provide TEM images (and/or SAED patterns) of synthesized graphene (by transferring it based on attaching a TEM grid and etching Cu), it should be very good. But, I know it is very difficult, so this comment is just a recommendation, not a mandatory task.”
Reply:
We appreciate the referee’s suggestion. TEM images or SAED patterns of synthesized graphene can visualize graphene crystalline. Actually, it is not easy to transfer the micro/nano synthesized graphene on TEM grid, which is too small to control. However, we provide some references about TEM images of irradiated PMMA sample as follows:
Figure (d) HRTEM images of the selected PMMA nanofiber area
“The SAED pattern in the Fig. (d) for the crossed part shows concentric rings, which indicates the structure was polycrystal-line.” (Duan H, et al. Turning PMMA Nanofibers into Graphene Nanoribbons by In Situ Electron Beam Irradiation. Adv. Mater., 2010:3284-3288.)
Figure (b) The HRTEM picture of the irradiated PMMA
“The crystallization of the negative PMMA film is firstly confirmed by the selected area electron diffraction (SAED) pattern, as shown in the Fig.(b).From the SAED pattern, several couples of symmetrical primary diffractive spots (as marked by the circle and square boxes) and lots of random high-order diffractive spots can be seen, which indicates that the fragment is really polycrystalline” (Chen W., et al. All-carbon based graphene field effect transistor with graphitic electrodes fabricated by e-beam direct writing on PMMA. Sci. Rep., 2015:12198)

Reviewer 2 Report
The paper can be accepted after major revisions:
Can the graphene nanostructures be transferred from the copper substrate? The copper substrate is useful for the graphene synthesis but it is not suitable for any possible applications.
Is the graphene synthetized by means of PMMA electron beam irradiation single-crystal or polycrystalline? If the material is polycrystalline the authors should provide the analysis for the D/G ratio for the size of the graphene grains.
Can the authors provide the Raman mapping of the areas reported in Figure 3 with a statistical evaluation of the graphene parameters as 2D width, D/G intensity ratio and the 2D/G intensity ratio?
In Figure 4c, the blue spectrum of the graphene after 10 minutes annealing at 900°C present the spectrum of a multilayer graphene with a asymmetric 2D peak, after 20 minutes the spectrum has the standard shape of a single or bilayer graphene. The author should expand the explanation of such figure and clarify better the mechanism of the decrease of the number of graphene layers with an increasing annealing time because the present explanation is obscure.
Author Response
Referee #2
Comments:
“The paper can be accepted after major revisions:”
Reply: We appreciate the referee’s positive comments on our work.
“1)Can the graphene nanostructures be transferred from the copper substrate? The copper substrate is useful for the graphene synthesis but it is not suitable for any possible applications.”
Reply:
Thanks for your review and we really appreciate for your time and patience. We have made further explanation as below:
The graphene layer formed on the metal film can be transferred to other substrate by using a sacrificial poly (methyl methacrylate) PMMA layer technology. Briefly, spin-coating a PMMA layer onto the sample, then the copper metal was etched away in FeCl3 or HCl solution and the detached film was placed in a water bath. After that, the film was transferred to target substrate, allowed to dry, and placed in an acetone bath to dissolve the PMMA support layer. After a rinse with isopropyl alcohol, the samples were characterized (Zheng M, et al. Metal-catalyzed crystallization of amorphous carbon to graphene. Applied Physics Letters, 2010, 96(6):183). However, it is not easy for graphene nanostructure transferring because micro/nano graphene is too small to control. Although the graphene nanostructure on copper substrate limited its application, we can try to selective etch copper substrate to construct some useful devices. We give out a specific application scheme to exhibit its potential application.
Figure. The schematic diagram of graphene vibrating sensor based on in-situ graphene nanostructure
“2) Is the graphene synthetized by means of PMMA electron beam irradiation single-crystal or polycrystalline? If the material is polycrystalline the authors should provide the analysis for the D/G ratio for the size of the graphene grains.”
Reply:
Thanks for your professional detailed revie. Commonly, G peak signifies the mono- or poly-crystalline graphite and the D peak reflects the defects and disordering of the graphite. The defect level of graphene can be determined by the D-to-G band intensity ratio, and low values of ID/IG suggest the formation of more homogenous and continuous graphene film (Jovana P. et al. Raman spectroscopy study of graphene thin films synthesized from solid precursor. Opt. Quant. Electron. 2016, 48:115). Then the graphene synthetized by means of PMMA e-beam irradiation is polycrystalline due to its obvious G peak and D peak in Raman test. Besides that, some literatures also show that graphene synthetized by means of PMMA electron beam irradiation is polycrystalline. The specific arguments are as follows:
Figure (d) HRTEM images of the selected PMMA nanofiber area
“The SAED pattern in the Fig. (d) for the crossed part shows concentric rings, which indicates the structure was polycrystal-line.” (Duan H, et al. Turning PMMA Nanofibers into Graphene Nanoribbons by In Situ Electron Beam Irradiation. Adv. Mater., 2010:3284-3288.)
Figure (b) The HRTEM picture of the irradiated PMMA
“The crystallization of the negative PMMA film is firstly confirmed by the selected area electron diffraction (SAED) pattern, as shown in the Fig.(b).From the SAED pattern, several couples of symmetrical primary diffractive spots (as marked by the circle and square boxes) and lots of random high-order diffractive spots can be seen, which indicates that the fragment is really polycrystalline” (Chen W., et al. All-carbon based graphene field effect transistor with graphitic electrodes fabricated by e-beam direct writing on PMMA. Sci. Rep., 2015:12198)
“3)Can the authors provide the Raman mapping of the areas reported in Figure 3 with a statistical evaluation of the graphene parameters as 2D width, D/G intensity ratio and the 2D/G intensity ratio?”
Reply:
Thanks for your professional advice. The Raman mapping for the area outline in figure 3(a) was added in the supplementary information. A series of statistical evaluation of the graphene parameters also have been presented to explain the mechanism of graphene synthesis.
Figure S1 (a) Raman mapping for the area outline in figure 3(a) at the 2D peak. (b) Raman spectra extracted randomly from Raman mapping area.
Just as shown in figure S1, it is the Raman mapping at 2D peak for the area outline in figure 3(a). There are some differences at different positions maybe result from the irregular change of the number of layers and defects of graphene because of copper particles agglomeration. In order to explain the mechanism of the phenomenon, the Raman spectra was extracted randomly from Raman mapping area for further discussion. The graphene parameters, including 2D width, D/G intensity ratio and the 2D/G intensity ratio, was calculated in table S1.
Table 1 a statistical evaluation of the graphene parameters from figure S1(b)
The 2D width of test areas in table 1 were varying from 37.7 cm-1 to 53.9 cm-1, which means a multilayer graphene sheet on copper surface. Commonly, the ID/IG ratio and I2D/IG ratio depends on the number of layers. ID/IG ratio (0.1~0.5) and I2D/IG ratio (0.5~1.4) also indicates that multilayer graphene sample was acquired on copper surface. Besides that, the ID/IG ratio of sample is smaller than 1.0, which implies the defects and disordering is also lower than other substrates.
“4) In Figure 4c, the blue spectrum of the graphene after 10 minutes annealing at 900°C present the spectrum of a multilayer graphene with an asymmetric 2D peak, after 20 minutes the spectrum has the standard shape of a single or bilayer graphene. The author should expand the explanation of such figure and clarify better the mechanism of the decrease of the number of graphene layers with an increasing annealing time because the present explanation is obscure.”
Reply:
Thanks for your professional and detailed review. Although high temperature annealing of graphene has been reported widely in preliminary study, it is rarely reported about the relationship between the number of graphene layers and annealing temperature. We have done our best to expand the explanation of mechanism of the decrease of the number of graphene layers with an increasing annealing time. The detailed information was added in results and discussion section (Figure 4c).
We added “There are two mechanisms to graphene synthesis on copper substrate in the annealing process: carbon precipitation and surface diffusion. The carbon atoms are firstly dissolved and precipitated on the copper surface, then some carbon atoms will be desorbed from the surface of metal layer under the low pressure ambient. As the result of limited carbon source, the number of graphene layers is gradually reduced with an increasing annealing time, which lead to a narrow and Gaussian distribution 2D peak in Raman test.” in results and discussion section (Zheng M, et al. Metal-catalyzed crystallization of amorphous carbon to graphene[J]. Appl. Phys. Lett., 2010, 96(6):183; Li X, et al. Evolution of graphene growth on Ni and Cu by carbon isotope labeling. Nano Lett, 2009, 9: 4268–4272;李汉超等. 退火时间对Ni催化非晶碳转变石墨烯的影响. 表面技术, 2019, v.48(06):80-86.).
We also added “26. Maxwell, Z.; Kuniharu, T.; Benjamin H., Hui F., Xiaobo Z., Metal-catalyzed crystallization of amorphous carbon to grpahene. Applied Physics Letters 2010, 96(10):1063.” in references section.

Round 2
Reviewer 2 Report
All the request are fully addressed and therefore the manuscirpt can be accepted in the present form.
This manuscript is a resubmission of an earlier submission. The following is a list of the peer review reports and author responses from that submission.
Round 1
Reviewer 1 Report
The paper can be accepted after major revisions:
Can the graphene nanostructures be transferred from the copper substrate? The copper substrate is useful for the graphene synthesis but it is not suitable for any possible applications.
Is the graphene synthetized by means of PMMA electron beam irradiation single-crystal or polycrystalline? If the material is polycrystalline the authors should provide the analysis for the D/G ratio for the size of the graphene grains.
Can the authors provide the Raman mapping of the areas reported in Figure 3 with a statistical evaluation of the graphene parameters as 2D width, D/G intensity ratio and the 2D/G intensity ratio?
In Figure 4c, the blue spectrum of the graphene after 10 minutes annealing at 900°C present the spectrum of a multilayer graphene with a asymmetric 2D peak, after 20 minutes the spectrum has the standard shape of a single or bilayer graphene. The author should expand the explanation of such figure and clarify better the mechanism of the decrease of the number of graphene layers with an increasing annealing time because the present explanation is obscure.
Reviewer 2 Report
This is an interesting work regarding the fabrication of nanostructure of graphene on copper substrates. The article is written to flow smoothly with the main take home messages to be pretty loud and clear. It is worthy of publication in order to elevate the existing knowledge within this field.
Some minor comments to be considered:
1) is it possible as future option to play with the graphite surface chemistry as for instance to have graphite oxide or other heteroatom?
2) is it possible another substrate to be used?
3) please consider regarding the synthesis of graphite the following approaches as presented in the article: 10.1039/D1TC01316E